Potential of artificial intelligence to accelerate diagnosis and drug discovery for COVID-19

Mikkili Indira 1 indu221007@gmail.com
Karlapudi Abraham Peele 1
Venkateswarulu T. C. 1
Kodali Vidya Prabhakar 2
Macamdas Deepika Sri Singh 1
Sreerama Krupanidhi 1
1 Biotechnology, Vignan’s Foundation for Science, Technology & Research , Guntur, Andhra Pradesh , India
2 Biotechnology, Vikrama Simhapuri University , Nellore, Andhra Pradesh , India
Orlov Yuriy
Electronic publication date: 2021 Oct 5
Publication date: 2021
Volume: 9
Electronic Location ID: e12073
Received 2020 Jul 31; Accepted 2021 Aug 5
Copyright: © 2021 Mikkili et al.
Copyright year: 2021
Copyright holder: Mikkili et al.
License: This is an open access article distributed under the terms of the Creative Commons Attribution License, which permits unrestricted use, distribution, reproduction and adaptation in any medium and for any purpose provided that it is properly attributed. For attribution, the original author(s), title, publication source (PeerJ) and either DOI or URL of the article must be cited.
License URL: https://creativecommons.org/licenses/by/4.0/

Keywords: Machine learning, Neural networks, Drug discovery, Reverse transcriptase polymerase chain reaction, Artificial intelligence, Computed tomography, SARS CoV-2, Pharmacogenomics, Homology modeling, Protein prediction

Funding: DST-FIST Networking Facility Project LSI-576/201 Vignan’s Foundation for Science, Technology and Research 522213 This work was supported by DST-FIST Networking Facility Project LSI-576/201 and Vignan’s Foundation for Science, Technology and Research (Vadlamudi-522213). The funders had no role in study design, data collection and analysis, decision to publish, or preparation of the manuscript.

==============================
The coronavirus disease (COVID-19) pandemic has caused havoc worldwide. The tests currently used to diagnose COVID-19 are based on real time reverse transcription polymerase chain reaction (RT-PCR), computed tomography medical imaging techniques and immunoassays. It takes 2 days to obtain results from the RT-PCR test and also shortage of test kits creating a requirement for alternate and rapid methods to accurately diagnose COVID-19. Application of artificial intelligence technologies such as the Internet of Things, machine learning tools and big data analysis to COVID-19 diagnosis could yield rapid and accurate results. The neural networks and machine learning tools can also be used to develop potential drug molecules. Pharmaceutical companies face challenges linked to the costs of drug molecules, research and development efforts, reduced efficiency of drugs, safety concerns and the conduct of clinical trials. In this review, relevant features of artificial intelligence and their potential applications in COVID-19 diagnosis and drug development are highlighted.

Introduction

The COVID-19 pandemic is a worldwide health crisis. The causative agent for COVID-19 disease is severe acute respiratory syndrome coronavirus 2 (SARS-CoV-2) (Singhal, 2020). Various vaccines and drugs are being tested in trials for the treatment of COVID-19 disease and released in to the market for emergency purpose. SARS-CoV-2 infects the respiratory tract, mainly the lungs that leads to acute respiratory syndrome and finally death in severely ill and co-morbid patients. One challenge facing the health care workers is identifying people with COVID-19 (Wu et al., 2020). The disease is diagnosed using the test reverse transcriptase polymerase chain reaction (RT-PCR), which is the preferred method. However, the time taken to run the test and return the results is 2 days. Furthermore, there is an under-supply of RT-PCR test kits, lack of advanced technologies limits the disease identification and drug development process (Tahamtan & Ardebili, 2020). An advanced and accurate diagnosis tool that yields rapid results is required (Carter et al., 2020). Artificial intelligence (AI) comprises advanced cognitive systems that are progressing rapidly across the world (Ahuja, 2019). AI algorithms can be used to enhance the findings from chest computed tomography (CT) imaging (Hosny et al., 2018) that enable the rapid diagnosis of a disease (Mei et al., 2020). Considering the public health perspective and the need to quickly isolate patients with COVID-19, chest CT is used for diagnostic imaging due to its sensitivity than chest radiography (Xu et al., 2020). There are few articles related to the detailed methods for detection of SARS CoV-2 and drug discovery using AI. AI tools and software can be used for the development of drugs in the pharmaceutical industry (Agrawal, 2018). Machine learning is a fundamental AI tool that uses algorithms to recognize various patterns in input data collected for disease as well as drug identification (Jiang et al., 2017). Within machine learning, the subfield of deep learning involves neural networks that play an important role in collecting the input data and generating the output data (Nguyen et al., 2019). Artificial neural networks include recurrent neural networks, multilayer perception and convolutional neural networks (Nguyen et al., 2019). A convolutional network consists of a dynamic system with local connections characterized by its biological system modeling, use of images, topology, processing of complex brain functions, recognitions, signal processing and video processing (Abiodun et al., 2018). Multilayer perception consists of pattern recognition, identification of process and controls in a single direction only (Abiodun et al., 2019). A recurrent neural network consists of a closed loop that memorizes and stores information (Abiodun et al., 2019). In the traditional drug discovery method, the first step is the target (disease) validation, followed by in vitro assays, screening of compounds, lead identification, preclinical trails, clinical trials (Phase I, II, III and IV) and finally approval of drugs for clinical use (Riaz et al., 2020). The entire process takes 10–15 years to develop a potential drug molecule for disease therapy (Mohs & Greig, 2017). With AI, drug discovery starts with prediction of a target protein’s role in disease, design of in silico compounds from libraries, novel target identification, prediction of structure–activity relationship, prediction of ADMET properties, drug repurposing, selection of patient population for clinical trials to increase success rates, pharmacovigilance, observation of adverse effects and interrogation of transcriptomic data (Paul et al., 2021). In every step in the development of potential lead molecules for disease therapy, AI plays an important role in minimizing the time required to produce good results (Bohr & Memarzadeh, 2020). Machine learning tools and software helps in identifying specific target virtual molecules and optimizing the efficacy and safety of the drug molecules in human population. AI reduces the time for identification of potential target molecules virtually compared with the synthesis of multiple compounds used for in vitro and in vivo assays (Paul et al., 2021). Currently, the health sector is facing the problems related to drugs and therapies for emerging diseases where nine out of ten drugs will fail in clinical trials and regulatory approval (Fleming, 2018). Using the AI technologies, the drug discovery process will speed up the drugs that are released into the market with short period of time, desired dose, overall safety, efficacy and other parameters required according to the individual patient need (Bender & Cortes-Ciriano, 2020; Paul et al., 2021).

The authors have identified the potential role of artificial intelligence in both disease and drug discovery process. The AI tools used for drug discovery process are highlighted in this manuscript. This article is intended for all researchers and academicians who studies about the applications of artificial intelligence related to medicine and health care. Considering the facts, the paper is arranged as follows. The review of literature is provided in a literature survey in which the SARS-CoV-2 detection methods, artificial intelligence role in disease identification and drug discovery are discussed. In another part, the features of artificial intelligence, the role of artificial intelligence for disease diagnosis and drug discovery and finally conclusions are addressed.

Survey methodology

The electronic searches were performed to retrieve literature in journal databases such as Google Scholar, Nature, WHO and Pub Med. A list of search terms can be seen in Table S1. The search terms used for collecting the data are SARS-CoV-2 disease, pathology, identification methods, RT-PCR, Immunoassay, artificial intelligence, drug discovery using artificial intelligence, machine learning, computed tomography, radiology, ophthalmology and COVID-19 disease data. The qualitative and quantitative articles were retrieved from literature and the insights related to COVID-19 disease diagnosis, measurable data to formulate the facts related to drug discovery aspects in research was highlighted. The following types of studies are included: review articles, research papers and short communications. The research papers with only abstracts, books and conference papers are excluded. The information provides the current knowledge by identifying the gaps in the literature, different theoretical perspectives and also suggests the future directions for research. All authors assessed the cited studies for quality of the information.

Literature survey

Mei et al. (2020) discussed how AI enabled diagnosis of COVID-19. In this study, AI algorithms were used to detect the disease based on chest findings, laboratory tests, exposure history and clinical symptoms. A total of 905 patients were tested using RT-PCR, and of them, 279 patients were diagnosed using AI. In 905 tested cases, 419 patients (46.3%) were positive for COVID-19 infection (Mei et al., 2020). AI system correctly identified 17 of 25 patients who had CT scans and tested positive for COVID-19 using RT-PCR. The researchers in this study highlighted the importance of AI to accelerate COVID-19 diagnosis (Mei et al., 2020).

Dhamad & Rhida (2020) provided insights about the serological and molecular methods for detection of COVID-19. In case of molecular methods RT-PCR, CRISPR-Cas 12 based method and isothermal amplification-based approaches are used for detection of COVID-19 where as in serological methods lateral flow assay and ELISA are used for detection of COVID-19. Molecular method RT-PCR is the gold standard method for detection of COVID-19 with accuracy.

Porte et al. (2021) evaluated the performance of two fluorescence immunoassays SOFIA SARS Antigen FIA and STANDARD F COVID-19 Antigen FIA. The two FIA kits showed 100% sensitivity and 96.9% specificity which indicates that the FIA kits can be used as diagnostic tools for early stages of COVID-19 infection other than RT-PCR.

A recent study by Vaishya et al. (2020) also highlighted the importance of AI and its applications for COVID-19. Eight different and significant applications for COVID-19 were discussed including early detection of disease, monitoring the treatment of patients, tracing the primary and secondary contacts of COVID-19-positive cases, projection of cases and mortality, drug development, vaccine development, reduction of workload on health care workers and finally disease prevention (Vaishya et al., 2020). This study provides support for the use of AI in detecting and managing disease in near future (Vaishya et al., 2020).

Li et al. (2020) evaluated the diagnostic accuracy of detecting COVID-19 disease and community-acquired pneumonia using pulmonary CT. In this study, deep learning technology was used to identify community-acquired COVID-19 on chest CT scans. This study used a multidisciplinary approach to diagnose the disease (Li et al., 2020).

Lalmuanawma, Hussain & Chhakchhuak (2020) discussed about AI and machine learning tools applications in COVID-19. The AI tools used for screening, prediction, contact tracing and drug development for COVID-19. Mohanty et al. (2020) highlighted the importance of drug repurposing for COVID-19 using AI. Deep learning algorithms can help in predicting potential antiviral drug molecules for COVID-19.

In a study reported by Stokes et al. (2020), machine learning techniques were used in drug discovery process. Deep neural networks are used for predicting the inhibitor molecule for disease causing agents. Although machine learning tools are available there is a need to discover and study the proteins of interest in disease and drug discovery process.

Allam, Dey & Jones (2020) reported on a health policy for detection of COVID-19 using an AI tool that can be applied internationally. In China, the companies Blue dot and Meta biota developed AI algorithms for accurate diagnosis of COVID-19. Ho (2020) discussed drug combinations for COVID-19, whose results must be evaluated to develop antiviral therapy for COVID-19. Drug combinations are screened and evaluated for drug dose compositions for personalized therapy are optimized using AI.

Das, Mishra & Gopalan (2020) highlighted the importance of open source machine learning tool for prediction of mortality risk among COVID-19 patients. The machine learning algorithm such as logistic regression, k-nearest neighbor, support vector machine, gradient boosting and random forest were used to analyze the mortality rate and found that logistic regression algorithm was the best one for mortality risk prediction.

A recent study by Kumar et al. (2021) highlighted the importance of artificial intelligence in diagnosis and prediction of COVID-19 disease. Further, the data sets and visualization techniques for COVID-19, designing of drugs for COVID-19 using artificial intelligence and role of deep learning and machine learning tools were discussed.

Features of ai

AI programming tools can help to identify signs of certain diseases (Caruso et al., 2020). Large volumes of health data are collected to identify a particular disease, including magnetic resonance images, X-ray results, CT scans, vaccinations, skin lesions, eye images, DNA sequences, blood samples, past medical history and current medications (Hosny et al., 2018) (Fig. 1). However, AI programming tools must be validated in large groups of people. Machine learning tools particularly deep learning algorithms have been used to diagnose disease automatically, which makes diagnosis cheaper and more accessible (Davenport & Kalakota, 2019). Using the data, the algorithms can draw conclusions in fraction of seconds and these can be accessed more easily for disease diagnosis. The process is inexpensive and less time-consuming for identifying disease compared with traditional methods of diagnosis (Agrebi & Larbi, 2020). In the case of COVID-19, two companies called Blue Dot and Meta biota (Allam, Dey & Jones, 2020) demonstrated the role of AI in early detection of SARS-CoV-2; Blue Dot predicted the outbreak areas in China, and Meta biota used big data analytics to count the outbreaks globally (Allam, Dey & Jones, 2020). In India, qure.ai algorithm tool has been used successfully to interpret the radiology images for diagnosis COVID-19and tuberculosis. The X-ray screening tool qXR (deep learning tool) was used for diagnostic and critical care management. Q Scout is another algorithm tool helps in contact monitoring of COVID-19 patients (https://qure.ai). RadVid-19 is the algorithm tool used by the Brazilian doctors to analyze chest X-rays and CT scans to find the spots in lungs which are markers of infection due to SARS-CoV-2 (https://radvid19.com.br/). Deep COVID-XR is the deep learning algorithm tool used for detecting COVID-19 in patients based on chest radiographs by US clinical radiologists during pandemic period (Wehbe et al., 2020). Elzeki et al., 2021 evaluated the performance of CXRVN (Chest X-ray COVID Network) deep learning computer aided model for early detection of COVID-19. In this model three different datasets were used and compared with pretrained models viz. Google net, Alex Net and Res Net. The evaluation results showed 94.5% accuracy for the proposed CXRVN model.

Figure 1 Illustration of how artificial intelligence uses the health data to identifying the disease.

The data related to CT scan, MRI, X-ray, skin lesions, eye, blood tests, patient medical history, medicines, vaccines and genetic information applied for diagnosing the disease using artificial intelligence (Source: https://mindthegraph.com).

Covid-19 detection methods

RT-PCR

Currently the corona virus SARS CoV-2 is detected using methods like RT-PCR, Rapid test kits, and CT scanning. In these methods, RT-PCR is preferred one and also called as gold standard method (Carter et al., 2020). In this method nasopharyngeal swab or oral pharyngeal swab is collected and kept in viral transport medium (Lopez et al., 2019). The genetic material i.e. RNA is extracted and converted into cDNA using reverse transcriptase enzyme. The cDNA further undergoes PCR amplification involves denaturation, annealing, extension and cleavage (Udugama et al., 2020). The DNA probe is used and in amplification process and it consists of reporter dye and quencher dye at 5′ end and 3′ end respectively. The reporter dye gets multiplied during amplification process, excited and leads to emission upon focusing a beam of light (fluorescence). The final step of this RT-PCR technique is conversion of light into digital data (Carter et al., 2020). The formation of graphs with cycle on x-axis and fluorescence intensity on y-axis is formed. The data is analyzed and considered as COVID positive or negative (Fig. 2). However, this method takes 2 days to release the output. In present adverse conditions there is undersupply of the RT-PCR kits. In order to renew this kit, something advanced and accurate tool are required whose response is efficient (Udugama et al., 2020).

Figure 2 Illustration of RT-PCR Assay protocol for detection of COVID-19.

The swab collected from suspected person is used for RNA extraction, conversion to cDNA, PCR amplification and finally fluorescence light data conversion into digital data. The digital graph describes the test sample is COVID-19 positive/negative. (Source: http://www.fredhutch.org/en/research/diseases/coronavirus/serology-testing.html; http://www.researchamerica.org/blog/explained-how-identify-active-covid-19-infection-people; http://www.medmastery.com/guide/covid-19-clinical-guide/how-rt-pcr-used-test-covid-19; www.advanced-biotechs.com/quantstudio).

Immunoassay method

The other method is immunoassay which is rapid test kit. In this, the immunoglobulins IgG and IgM antibodies are measured against the SARS-COV-2 proteins. The method is considered as fast and precise as it produces output in 10–15 min by examining whole blood (Jacofsky, Jacofsky & Jacofsky, 2020). In this the blood sample is collected and tested in which red colored lines are formed opposite to their respective sites as shown in Fig. 3. The presence of IgG and IgM antibodies results in the formation of red colored lines against ‘G’ site and ‘M’ site respectively (Jacofsky, Jacofsky & Jacofsky, 2020). Formation of red lines at the minimum of two confirms that the sample is having SARS-COV-2 proteins. However, this technique is not specific as it cannot detect virus in the incubation period and may show false negative result which leads to adverse effect in future (Udugama et al., 2020).

Figure 3 Immunoassay detection of COVID-19 disease.

Ct and ai

Computed tomography (CT) is a sensitive medical imaging technique used for identification of diseases. It involves imaging the organs with the help of computational knowledge (Shi et al., 2020). AI algorithms are used to enhance the chest CT findings for rapid diagnosis. The method is sensitive as it captures every small detail of the lungs. For detection of COVID-19, the imaging features used are ground glass opacities, consolidations and crazy paving patterns (Ozturk et al., 2020). Considering the public health perspective and the need to quickly isolate patients with COVID-19, chest CT is used because it is more sensitive than chest radiography and other techniques. However, these methods can be used in combination for accurate and better results and can be accelerated with new AI algorithms (Xu et al., 2020). AI accelerates the process compared to conventional techniques and does not involve symptomatic treatment. Daily updates from patients can be studied, and the rate of infection can be inspected by observing each movement of an infected person (Davenport & Kalakota, 2019). A recent study by Li & Xia (2020) reported that CT scan data was analyzed for 51 patients and found good results. Hence, CT scan data can be used for rapid diagnosis of COVID-19. The AI-based mobile app AI4COVID-19 was developed for evaluating COVID-19 patients based on cough samples (Imran et al., 2020).

Artificial intelligence in drug discovery

The use of AI in pharmacogenomics and drug development quickens the process of drug production. Drug development and market release takes at least 14 years, but AI uses an in-silico approach, which makes in vitro and in vivo tests easier (Zhavoronkov, Vanhaelen & Oprea, 2020). The first step in drug discovery using AI is to set a hypothesis. Next the hypothesis is validated by selecting potential lead molecules from the lead database. The activity of the lead molecules is evaluated using in-vitro and in-vivo tests (Vamathevan et al., 2019). The potential lead molecule is said to be a drug after optimizing parameters like affinity to bind the target, adverse effects, efficacy, bioavailability and therapeutic effect (Batool, Ahmad & Choi, 2019; Chan et al., 2019). The potential lead molecule from thousands of compounds is identified using neural networks, homology modeling, protein fold prediction, omics analytics, SMILES, LSTM models, etc. (Fig. 4) (Yang et al., 2019). The drug toxicity is essential to prevent the side effects and toxic effects in humans. The web-based tools used for evaluating the toxicity are Deep Tox, Lim Tox, Tox21 and admetSAR (Cheng et al., 2012; Yang et al., 2018). In COVID-19 disease, the enzyme 3CLpro can be considered as a target that is inhibited by a lead molecule. D. novo design tools would be used to obtain a potential drug candidate for COVID-19 (Amin et al., 2020). The AI tools used in drug discovery are PADME, MANTRA, PREDICT, deep neural Net QSAR, Deep chem, Deep Tox, XenoSite, SMARTCyp, FAME, ORGANIC, Potential Net, Hit Dexter, Delta Viva, Neural graph finger print, Alpha fold and Chemputer (Paul et al., 2021).

Figure 4 Steps involved in artificial intelligence for drug development.

However, advanced and new algorithms are required to substitute the human brains in near future. Collaboration among biotechnology, pharmaceutical and AI companies could streamline research and development efforts to identify novel, rare drug molecules and also personalized medicine.

Conclusion

The AI supports the future needs of medicine and health care by accelerating disease identification and drug discovery process. The technology develops the machine learning tools in health care particularly in the area of emerging diseases like COVID-19, regenerative medicine, gene therapy and pharmacology. Further, the artificial intelligence entails personalized medicine, assisted diagnostics, biomarkers, drug discovery and development. However, the advanced and new algorithms are required to substitute the human brains in near future. This enables an opportunity to collaborate biotech, pharma and artificial intelligence companies to streamline their research and development to identify the novel, rare drug molecules and also personalized medicine.

Supplemental Information

Supplemental Information 1 Search strategy for literature review.

Click here for additional data file.

The authors thank mind the graph for providing the web platform to draw the figure.

Additional Information and Declarations

Competing Interests

Author Contributions

Data Availability

The authors declare that they have no competing interest.

Indira Mikkili, Abraham Peele Karlapudi and Deepika Sri Singh Macamdas designed manuscript and figures.

T. C. Venkateswarulu, Vidya Prabhakar Kodali and Krupanidhi Sreerama analyzed the data, reviewed and approved the draft.

The following information was supplied regarding data availability:

This is a literature review therefore there is no code or raw data.

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
