# Peer review of "Potential of artificial intelligence to accelerate diagnosis and drug discovery for COVID-19"

_PeerJ, doi:10.7717/peerj.12073_

## Round 0.1 · original submission · Major Revisions

The manuscript got some critical remarks from the reviewers. Please check carefully the comments from review #1. Despite the importance of the problem of COVID-19 research, the work should have its value - what is new in AI applications here? Could it be presented in a systematic view for COVID-19? The manuscript style should correspond to the review article standards. Please add recent literature, as suggested by the reviewers. I encourage fast resubmission of the revised manuscript.

·

Basic reporting

The author reviewed this MS by how Artificial Intelligence can be utilized not only in disease identification but also in drug discovery. However, your article is inadequately presented. Besides, there are numerous grammatical mistakes and spelling mistakes, as well.
Although the article has scientific rigor, several major issues need to be changed before publication.

Experimental design

1. The current manuscript will advance no new information/recommendation for the readers.
2. Many non-scientific and incorrect/wrong information/sentences are there, which may mislead the readers.
3. The authors just have addressed several points randomly. Many sentences/information throughout the manuscript have serious flaws that withdrawn my thought from it.
4. Every section of the manuscript must be written scientifically according to the possible literature with relevant references.
5. The language needs to be improved considerably, mostly in the introduction and abstract. There are several errors that make it difficult to understand the manuscript.
6. The work seems preliminary. The study problem is not apparent. Can the authors explain the importance of Artificial Intelligence in the introduction section?
7. The article title is very vague. The authors need to simplify the title. It looks like a research MS by sighted the title.
8. The authors make errors in the naming of COVID-19 and SARS-CoV-2. SARS-CoV-2 is a virus that causes COVID-19 (Coronavirus Disease-19). Please make sure that this is correct.
9. The abstract section is inacceptable—no focus point in the abstract section.
10. The introduction section is inapplicable. Need to change the introduction considerably. Try to include the existing barriers also, how the present review unties those limits.
11. English is feeble. The authors need to improve their writing style. The whole manuscript needs to be checked by native English speakers.
12. Authors are suggested to replace \'Covid-19\' with \'COVID-19\' throughout the manuscript.
13. The research questions could be defined more precisely. This is probably due to language concerns.
14. The author should discuss more literature survey with references.
15. What is the need for 'COVID-19’ detection methods?

Validity of the findings

1. ‘Artificial Intelligence in disease identification and drug discovery’ should be the main focus point. The authors need to elaborate on this section in detail.
2. Very few references are cited here. This is a review article. The authors should increase the number of references.
3. The conclusion needs to address future perspectives.

Additional comments

Conclusively, there are likewise problems to address. I found this manuscript very foundation and do not recommend to publish in the present form. I also find the present manuscript lacks for its originality. Fundamental changes are obligatory before go for the final version.

Reviewer 2 ·

Basic reporting

The present review targeted potential area that is vital in disseminating the role of AI in COVID-19 patients data in disease identification and drug discovery approach. However, authors have not gone in depth of machine learning, IoT and role big data analytics in a way that meets the standard of PeerJ guidelines. The following drawbacks are noted
1. lack of professional English
2. repeated use of similar references (Literature Survey)
3. Though, authors tried to pull literature from latest published literature, there is scope for improvement in structure of article.

Experimental design

The study design is very much focused on survey of recent literature and lacks coverage from global resources (databases) like WHO on COVID -19 from UK, US, Brazil, INDIA which are worst hit and how they are doing data analytics using Big data and machine learning. There is vast amount of literature recently deposited globally on the subject under discussion in the review paper. This is major drawback of the review paper where it lacks professional approach.

Validity of the findings

Though, authors have developed and supported argument that meets the goals set out in the introduction. However, authors have not focused on role of AI in drug discovery in detail. this leads to inconsistency in the whole approach of Review paper.

Conclusion identifies future scope of the work on AI.

Additional comments

The review sheds light of AI and drug discovery which is critical area of research in COVID-19 in diagnosing disease and control authors have a not fully met the standards of PeerJ journal which is having wide audience.

Reviewer 3 ·

Basic reporting

The article is well written. The content included is quite good and is useful to the scientific community working for COVID-19. The concepts are well mentioned and the manuscript is also organized neatly.
However, the references mentioned in the manuscript are not in the same format. Please change the references to this journal format.

Experimental design

no comment

Validity of the findings

no comment

Additional comments

To conclude on this article, change all the references as per this journal format.

---

## Round 0.2 · Minor Revisions

Thanks for the manuscript update. The reviewer still has critical comments.

In addition to general reviewing remarks, I have some editorial advice.

Please add a section about AI and drug discovery to the text. The number of publications on COVID-19 topic is growing. Please check recent papers focusing on AI approaches to drug discovery to make your work more focused on the topic.

·

Basic reporting

The author reviewed this manuscript by how Artificial Intelligence can be utilized not only in disease identification but also in drug discovery. However, your article is inadequately presented.
Although the article has scientific rigor, several minor issues need to be changed before publication.

Experimental design

1. The language still needs to be improved, mostly in the introduction section. There are several errors that make it difficult to understand the manuscript.
2. The introduction section is inapplicable. Need to change the introduction considerably. Try to include the existing barriers also, how the present review unties those limits.
3. Please add a section named AI IN DRUG DISCOVERY.

Validity of the findings

1. Please add a section named AI IN DRUG DISCOVERY.
2. Figures 1 and 2 need to illustrative, not just presenting in a text form.

Additional comments

Though the authors addressed some of my comments still, I found this manuscript not suitable for publication. Minor changes are obligatory before go for the final version.

---

## Round 0.3 · Minor Revisions

Thanks for the manuscript update. There are some minor remarks on the references and the figure presentation. See the remarks from reviewer #1. The topic on COVID-19 drug search is very competitive, so, the paper has to be of high quality including formatting and presentation style. Please check recent publications including recent papers at PeerJ to refer to the latest findings in this area. Waiting for resubmission of the updated manuscript.

·

Basic reporting

Title: Potential of artificial intelligence to accelerate 1 diagnosis and drug discovery for COVID-19 (#51371)

Journal: PeerJ

The author reviewed this manuscript by how Artificial Intelligence can be utilized not only in disease identification but also in drug discovery. However, your article is inadequately presented.
Although the article has scientific rigor, several minor issues need to be changed before publication.

Experimental design

1. Check the references. Many inappropriate (format and style) are in there.
2. Check the Figure 1 legend.

Validity of the findings

1. Figures 2 need to illustrative, not just presenting in a text form.

Additional comments

Title: Potential of artificial intelligence to accelerate 1 diagnosis and drug discovery for COVID-19 (#51371)
Journal: PeerJ

BASIC REPORTING

The author reviewed this manuscript by how Artificial Intelligence can be utilized not only in disease identification but also in drug discovery. However, your article is inadequately presented.
Although the article has scientific rigor, several minor issues need to be changed before publication.

EXPERIMENTAL DESIGN

1. Check the references. Many inappropriate (format and style) are in there.
2. Check the Figure 1 legend.

VALIDITY OF THE FINDINGS

1. Figures 2 need to illustrative, not just presenting in a text form.

Reviewer 2 ·

Basic reporting

After revision, manuscript looks professional with literature.

Experimental design

Article content is within the Aims and Scope of the journal.

Validity of the findings

The conclusion address future needs of medicine and health and supports with recommendation.

---

## Round 0.4 · accepted · Accept

Sorry for the multiple revision rounds. The research topic is very important, but we have to check manuscripts. The reviewers have no more remarks. I recommend publishing the manuscript now.

·

Basic reporting

The author reviewed this manuscript by how Artificial Intelligence can be utilized not only in disease identification but also in drug discovery. However, your article is inadequately presented.
The authors addressed my comments accordingly. Just some minor spell checks are required (corona virus will be coronavirus, etc.).
The revisions look good to me. The manuscript is good enough to publish in the PeerJ journal.
The authors did an excellent job.
Congratulations!!!

Experimental design

The revisions look good to me. The manuscript is good enough to publish in the PeerJ journal.

Validity of the findings

The revisions look good to me.

Additional comments

Looks okay now.